# Relationship between anxiety and internet searches before percutaneous ultrasound-guided diagnostic procedures: A prospective cohort study

Marcio Meira[1,2]*, Almir Galvão Vieira Bitencourt[1], Demian Jungklaus Travesso[1], Rubens Chojniak[1], Paula Nicole Vieira Pinto Barbosa[1]

1 Department of Interventional Radiology, AC Camargo Cancer Center, Radiology Department, São Paulo, Brazil, 2 Department of Interventional Radiology, University of Sao Paulo Medical School, São Paulo, Brazil

☯ These authors contributed equally to this work.
* marciomeira2050@gmail.com

**Data Availability Statement:** All relevant data are within the paper and its Supporting Information files.

## Abstract

Invasive procedures guided by ultrasound (US) are part of routine medical diagnostic investigation. The lack of knowledge surrounding the technical aspects of such procedures can lead patients to seek complementary information on the Internet, which may in turn trigger anxiety. However, the intersection between the fields of Radiology and Psychology is poorly studied. Here, we identify the profile of an anxious patient before an US-guided intervention. We prospectively studied 133 patients undergoing image-guided procedures. The State-Trait Anxiety Inventory (STAI) was applied for psychometry. Significantly higher anxiety scores were observed in female patients ($p = .001$), those who believed they had received inadequate information from their referring physician ($p = .006$), and in patients who considered online information unreliable or difficult to access ($p = .007$ and $p = .001$, respectively). Participants who defined themselves as proactive online reported lower anxiety levels ($p = .003$).

## Introduction

Ultrasound (US)-guided invasive procedures are essential modalities in clinical oncology. Advantages of US techniques include real-time imaging, rapid results, low costs, portability, safety and radiation exposure avoidance [1]. Technological advances in medical imaging devices have given rise to an expanding set of interventional procedures designed to benefit the increasing patient population [2]. In parallel, the Internet is increasingly utilized to retrieve information related to medical procedures prior to communication with a healthcare professional [3]. However, online information can be misleading and may amplify misperceptions rather than provide clarification [4]. Given the large amount of inaccurate information online, people can easily become misinformed [5]. While there is a greater awareness of the need to minimize anxiety levels before image-guided procedures [6], very few studies correlate these interventions with the use of the Internet as a consultation tool.

**Funding:** The author(s) received no specific funding for this work.

**Competing interests:** he authors have declared that no competing interests exist.

Studies that evaluate Internet usage by patients in search of health information show conflicting data. Similarly, results from studies that assess Internet usage by health professionals are divergent. Several researchers characterize the Internet as both a therapeutic tool [7] and a device to improve doctor-patient relationships. A study conducted by Google® found an increasing rate of health-related searches amongst its users, with a higher number of respondents having sought the platform as their first source of information (35%) than those who immediately consulted a physician (26%) [3]. Another survey conducted by the same platform [8] revealed the primary topic participants searched for online was medical treatment (60%), followed by general information about diseases (52%) and symptoms (48%). However, uncertainties and difficulties in comprehending medical language are often reported due to the multiplicity of information available online [9].

The intersection between the fields of Radiology and Psychology is poorly studied. Considering radiology tests are conducted in the initial approach for diagnosing and monitoring of cancer treatment, it is extremely important that this intersection be further studied. Therefore, the objective of the present study was to investigate the demographic and psychological profiles of patients undergoing US-guided invasive procedures, the patients' online searches about their proposed interventions, and the impact these variables had on their anxiety levels before the procedure.

## Methods

### Study design

We conducted a prospective study of patients undergoing US-guided invasive procedures at a cancer center. This study was approved by our Institutional Review Board and performed in accordance with the ethical standards outlined in the Declaration of Helsinki (certificate number approval by the ethics committee 66022517.8.0000.5432, appraisal 2.063.731). Between July 2018 and July 2019, patients were recruited for participation while in the waiting room for their image-guided invasive procedure. All patients that were approached agreed to participate in the study and provided informed consent before inclusion.

Fig 1 depicts the criteria used in patient selection. Hospitalized patients were excluded from the study due to time restrictions before medical interventions.

Patients were evaluated at two separate time points. First, prior to the procedure patients completed the State-Trait Anxiety Inventory (STAI). This self-administered questionnaire has been used since 1970 [10] and has the most robust validation in medical literature. It measures state and trait components of anxiety with questions relating to the present moment. After the procedure, patients completed another self-administered questionnaire that addressed multiple variables including education, socio-demographics, and pain levels during the procedure, as well as subjective questions concerning the reliability of information available on the Internet. Notably, participants were informed that there were no 'correct' answers in either of the questionnaires. Fig 2 outlines the steps taken from the patients' arrival at the hospital to the application of the questionnaires and the performance of the procedures.

### Pre-procedure questionnaire

STAI consists of two subdivisions, each with unique characteristics. The first is the Anxiety-State scale (STAI-State), which assesses states of anxiety in the present moment by asking respondents how they feel "now," using items that measure subjective feelings of apprehension, tension, nervousness, and concern. The second is the Anxiety-Trait scale (STAI-Trait), which assesses personality characteristics related to anxiety propensity, including general states of calmness, confidence, and security. This psychometric tool is self-administered and consists of 40 items, with 20 allocated to each subscale. The scores range from 20 to 80 in both subsets,

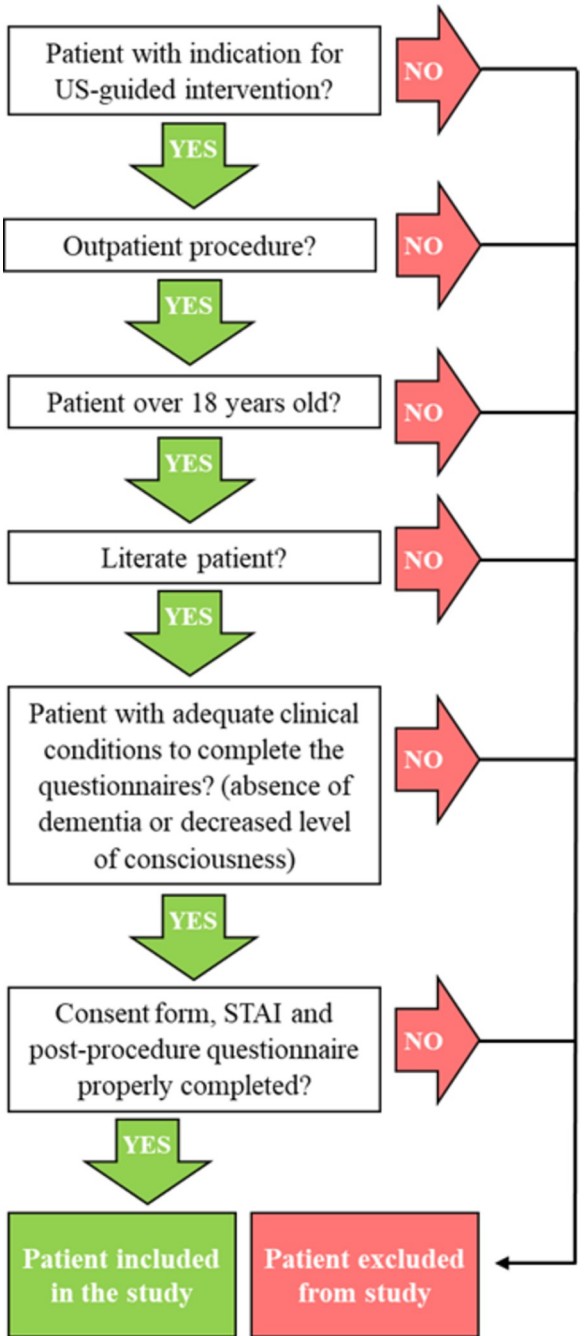

**Fig 1. Inclusion criteria.** STAI, State -Trait Anxiety Inventory; US, ultrasound.

with higher scores indicating greater anxiety levels. Generally, a cutoff value of 39–40 points is used to detect symptoms of clinically significant anxiety [10–13]. Most adults require approximately 10 minutes to complete the questionnaire.

## Post-procedure questionnaire (supplementary material)

The following clinical and sociodemographic data were collected: age, gender, education level and patient referral source (public or private health care). Patients were asked to provide their

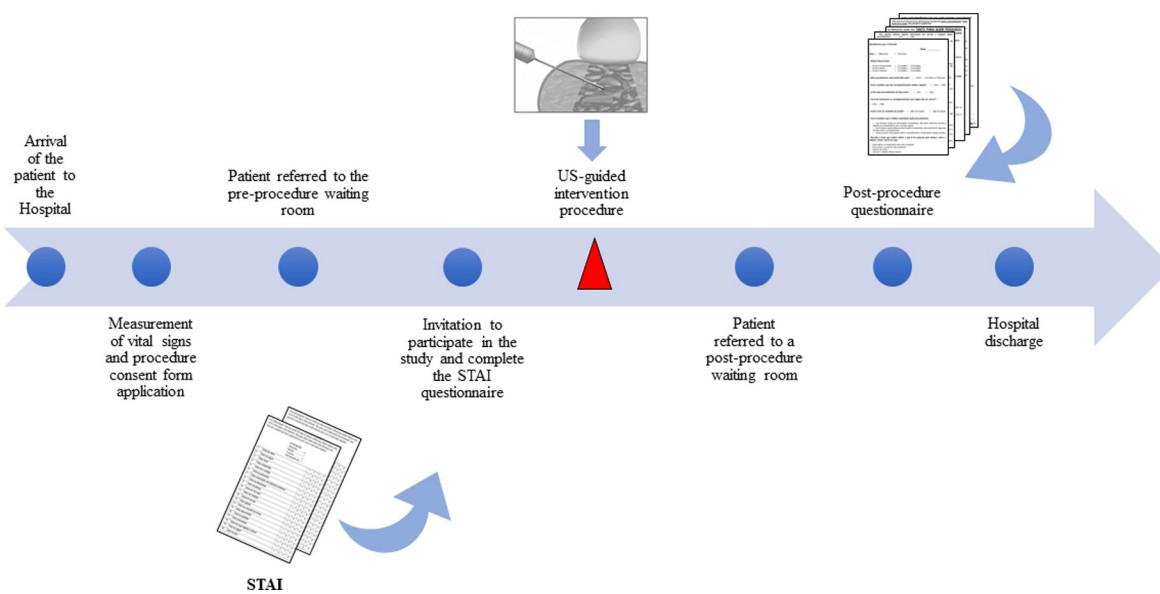

**Fig 2. Flowchart representing the steps taken from the patients' arrival at the hospital to the presentation of the questionnaires and the performance of the procedures.** STAI, State -Trait Anxiety Inventory; US, ultrasound.

medical history and gave responses to subjective questions regarding regular medical follow-ups and overall health (whether they considered themselves "healthy" or "unhealthy").

Closed questions were used for respondents to evaluate the amount of information provided to them by the referring physician before their procedure, as well as their proactivity in searching for medical information online. The subjective concept of proactivity was assessed by asking, "A proactive person is one who seeks to anticipate problems; to foresee situations... Do you consider yourself to be proactive when searching for information about your health?" Patients were also asked to appraise the reliability and accessibility of online information.

## Statistical methods

The STAI-State and STAI-Trait anxiety scores were compared with other variables. Normality distribution was tested using the Kolmogorov-Smirnov test, with p-values of .021 for State and .004 for Trait obtained. The non-parametric Mann-Whitney test was used to compare two subgroups, and the Kruskal-Wallis test was used when three subgroups were analyzed. In the 3-group comparison, whenever statistically significant differences were detected, the Kruskal-Wallis multiple comparison test was used. Scores of $p < .05$ were considered statistically significant. SPSS software was used (version 20.0).

## Results

133 patients underwent US-guided invasive procedures. The mean age of respondents was 49.7 years. Procedures comprised fine needle aspiration (FNA) and core needle biopsy (CNB). 77 patients (57.8%) underwent FNAs at anatomic sites, distributed as follows: 50 thyroid, 10 breast, nine axillary lymph nodes, four cervical lymph nodes, three parotid, and one soft tissue lesion. 56 patients (42.2%) patients underwent core biopsies: 50 breast, three prostate, and three cervical lymph nodes. Baseline characteristics are shown in Table 1.

Female respondents reported higher mean levels of anxiety than male respondents in the STAI-State scale ($p = 0.001$) (Table 2). There was a statistically significant inverse relationship

**Table 1. Baseline characteristics of patients undergoing US-guided invasive procedures.**

| Parameter | US | |
|---|---|---|
| **Sex** | **n** | **%** |
| Female | 109 | 82% |
| Male | 24 | 18% |
| **Education** | | |
| Incomplete Fundamental | 6 | 4.5% |
| Complete Fundamental (9 years) | 6 | 4.5% |
| Incomplete High School | 5 | 3.8% |
| Complete High School (12 years) | 22 | 16.5% |
| Incomplete University education | 25 | 18.0% |
| Complete University education ($\geq$ 14 years) | 69 | 51.9% |
| **Origin** | | |
| Public Health System | 29 | 21.8% |
| Private Health System | 104 | 78.2% |
| **Regular medical attention[1]** | | |
| Yes | 120 | 90.8% |
| No | 13 | 9.2% |
| **Procedure previously performed** | | |
| Yes | 55 | 41.4% |
| No | 78 | 58.6% |
| **Previous cancer treatment** | | |
| Yes | 45 | 33.8% |
| No | 88 | 66.2% |
| **Individual subjective aspect of health status [2]** | | |
| Healthy | 127 | 95.5% |
| Unhealthy | 6 | 4.5% |

US, ultrasonography.

[1] Patients asked if they considered themselves to have regular medical follow-up.

[2] Patients asked about their perception of their general health.

between patient assessment of the amount of information received from the referring physician and STAI-State anxiety scores ($p$ = 0.006) (Table 3). As this parameter had three possible responses, the Mann-Whitney test was used to compare responses two by two. Statistical significance was found in the comparisons between the responses "limited information" and "all necessary information," and between "almost all necessary information" and "all necessary information" ($p < 0.05$). The relationship between the patients' self-assessed proactivity and STAI scores are shown in Table 4.

**Table 2. Comparison of anxiety (assessed by STAI) between genders.**

| STAI | Sex | Mean | Median | SD | n | IC | *P* |
|---|---|---|---|---|---|---|---|
| **State** | Female | 44.0 | 45.0 | 9.1 | 109.0 | 1.7 | 0.001 |
| | Male | 36.6 | 37.0 | 8.4 | 24.0 | 3.4 | |
| **Trait** | Female | 39.7 | 39.0 | 8.2 | 109.0 | 1.5 | 0.063 |
| | Male | 36.2 | 34.5 | 9.8 | 24.0 | 3.9 | |

STAI, State -Trait Anxiety Inventory; US, ultrasound.

**Table 3. Comparison between anxiety levels (assessed by STAI) and information previously provided by the referring physician.** Respondents were asked the following question: "Do you believe that the referring doctor provided all the necessary information to you prior to your procedure?".

| STAI | Answer | Mean | Median | SD | N | IC | *p* |
|------|--------|------|--------|----|----|----|-----|
| **State** | Limited information | 49.2 | 48 | 6.6 | 5 | 5.8 | 0.006 |
| | Almost all necessary information | 47.3 | 46 | 8.7 | 22 | 3.6 | |
| | All necessary information | 41.2 | 42 | 9.2 | 105 | 1.8 | |
| **Trait** | Limited information | 49.4 | 55 | 12.6 | 5 | 11.0 | 0.075 |
| | Almost all necessary information | 41.0 | 40,5 | 8.4 | 22 | 3.5 | |
| | All necessary information | 38.1 | 38 | 8.1 | 105 | 1.5 | |

STAI, State -Trait Anxiety Inventory.

There was a significant correlation between STAI-State scores and how reliable patients found information on the internet concerning their procedure, with higher scores associated with "unreliable" responses (*p* = 0.007). STAI-State scores were also directly related to assessments of the accessibility of online information (mean of 39.3 points for those who considered accessibility "easy", compared to 45.4 points for those who reported it to be "difficult" (*p* = 0.001), Table 5. There were no significant associations between STAI anxiety scores and age, education level, past experiences of undergoing the proposed procedure or cancer treatment, subjective assessments of health status, or the regularity of medical care (*p* > 0.05).

## Discussion

The profile of a patient who undergoes an US-guided percutaneous interventional procedure is likely a female who reports regular medical care and has higher education and private health insurance. This patient undergoes the procedure for the first time, does not receive cancer treatment, and believes they are healthy. We observed significantly higher anxiety scores among female patients, respondents who reported to have had received insufficient information from their referring physician, those who considered information available on the Internet unreliable, and participants who found it difficult to access information online. On the other hand, patients who defined themselves as proactive in their online searches demonstrated lower anxiety levels before invasive interventions. These results are relevant in clinical practice because there are few studies that associate Internet use, information communicated by referring physicians, and anxiety levels prior to US-guided invasive procedures.

Our results show that females presented higher levels of anxiety than males, which is consistent with the findings of several published studies. Yu et al. [11] conducted a study on cancer patients before diagnostic imaging exams. Of the 328 participants, 152 (46.3%) reported having anxiety and females were found to have higher levels of anxiety than males (*p* = 0.021). Surgical studies are more prevalent in medical literature. Despite inherent differences between

**Table 4. Comparison between anxiety levels (assessed by STAI) proactivity.**

| STAI | Answer | Mean | Median | SD | n | IC | *p* |
|------|--------|------|--------|----|----|----|-----|
| **State** | Yes | 41,8 | 42 | 9.6 | 108 | 1.8 | 0.003 |
| | No | 47,4 | 48 | 5.8 | 23 | 2.4 | |
| **Trait** | Yes | 38,6 | 38 | 9.0 | 108 | 1.7 | 0.129 |
| | No | 41,0 | 40 | 6.4 | 23 | 2.6 | |

STAI, State -Trait Anxiety Inventory.

**Table 5. Comparison between anxiety levels (assessed by STAI) and patients' subjective appraisals of the reliability and accessibility of online information regarding US-guided invasive procedures.** The available data refers to both those who did and did not search online. Anxiety scores were measured using the STAI scale.

| STAI | Answer | Mean | Median | SD | N | IC | p |
|---|---|---|---|---|---|---|---|
| **State** | Reliable | 40.2 | 40.5 | 10.6 | 60 | 2.7 | 0.007 |
| | Unreliable | 45.0 | 45.0 | 7.3 | 58 | 1.9 | |
| **Trait** | Reliable | 38.1 | 36.5 | 9.1 | 60 | 2.3 | 0.119 |
| | Unreliable | 40.2 | 39.5 | 8.3 | 58 | 2.1 | |
| **State** | Easy | 39.3 | 40 | 10.2 | 55 | 2.7 | 0.001 |
| | Difficult | 45.4 | 45 | 7.5 | 65 | 1.8 | |
| **Trait** | Easy | 37.5 | 37 | 8.2 | 55 | 2.2 | 0.060 |
| | Difficult | 40.5 | 39 | 8.9 | 65 | 2.2 | |

STAI, State -Trait Anxiety Inventory.

populations, such studies can be instructive when examining the difference in anxiety levels between genders. Domar et al. [12] evaluated 523 patients undergoing elective surgery. Preoperatively, multiple parameters were evaluated including patients' age, gender, occupation, education level, type of surgery, and whether they had previously undergone a similar procedure. Respondents completed the STAI questionnaire for anxiety psychometry in the waiting room right before their intervention. Of all the parameters examined, only gender was found to be positively correlated with anxiety levels, with females presenting as more anxious. This finding has also been reported in other parts of the world [13]. Jafar et al. used STAI to evaluate 300 pre-surgical patients in Pakistan and found higher levels of anxiety in female respondents. However, considering anxiety questionnaires are self-administered, female patients may be more inclined to admit to having anxiety than their male counterparts. It is therefore necessary to exercise caution before stating that these data represent an innate difference in anxiety levels between genders [14].

Patients who claimed to have received insufficient information from referring physicians were found to be more anxious before US-guided interventions. The lack of knowledge amongst patients prior to undergoing medical interventions has been the subject of previous studies. Kiyohara et al. [15] evaluated 140 patients before undergoing elective surgical procedures. STAI scores were correlated to the patients' understanding of their diagnoses, surgical procedures, and types of anesthesia. Knowledge about the diagnosis or the prescribed anesthesia did not significantly influence anxiety levels amongst participants. However, patients who had doubts about their surgical procedure reported higher STAI-State anxiety psychometry scores. Our results indicate patients with limited prior medical information have higher STAI-State scores, which supports previous studies that highlight the importance of patients receiving adequate information before medical procedures.

Various studies have been conducted to elucidate patient behavior after medical appointments. Bell et al. [16] used an online questionnaire to evaluate 274 members of an online community who had undergone a medical consultation within the 30 days prior to the study. Most respondents reported having searched for information online after their appointments (68%). Those who reported that they received insufficient information in their consultation were more likely to search online. Li et al. [17] evaluated 311 patients who had medical consultations in 2019. Participants listed curiosity and the perception that physicians gave incomplete information as the primary reasons for searching for complementary data on the Internet. There are, however, some limitations in these studies that are important to note. Firstly, they were limited to members of small online communities, raising questions regarding the generalization of their findings. In addition, as participants were connected to the Internet, it is

possible that they were more familiar with this technology than the general population. On the other hand, the present study brings new information to medical literature, as it included patients in the waiting room right before their procedures. Further, the questionnaires were applied regardless of whether participants had searched for complementary information online. Despite these methodological differences, we believe that our results are complementary and indicate that misinformation may lead patients to search for supplementary information online.

Patient appraisal of online information was also assessed. Both participants who assessed online information as "unreliable" and those who reported difficulty with accessibility were found to have higher anxiety levels. Previous studies have attempted to identify which criteria are credible quality indicators of online information. Johnson et al. [18] concluded that the formation of trust is influenced both by central parameters such as website content, as well as peripheral factors such as style and ease of access. On the contrary, Kelton et al. inferred that reliability assessments are linked to users' personal concepts and their identification with the available content. They proposed that a sense of concurrence between the user and the website results from the commonality between the information presented and "the user's own sense of identity, goals and values." [19]. Consequently, personal identification is seen to play a central role in creating a perception of "reliable" information. Therefore, we have identified a clinical interaction amenable to intervention, in which a more transparent communication interface between online information and the patient could decrease anxiety before an US-guided invasive procedure. A potential example would be the recommendation of trusted sites by physicians during consultations, preferably in the institution where the procedure takes place.

Proactivity in initiating online searches for information relating to health has also been studied. Murray et al. [20] conducted a telephone survey that covered all regions of the United States and consisted of 3209 participants. "Proactive" individuals were defined as those who instigated an online search for health topics of their own accord. The author concluded that proactive participants were more likely to consider themselves as "excellent" or "particularly good" at assessing the reliability of online information, as well as in their ability to find relevant information. The present study demonstrated that patients who were more proactive were less anxious at the time of their US-guided interventions. Mc Mullan et al. [21] published a literature review on the use of the Internet to obtain health information and its impact on doctor-patient relationships. Three potential responses to patients who utilize online searches prior to consultation were discussed: 1) the healthcare professional feels threatened by the information and responds by defensively stating his "expert opinion"; 2) the healthcare professional and the patient cooperate in the analysis of online information; 3) the healthcare professional guides patients to reliable sites on the Internet. Previous studies suggest positive results when information found on the Internet by the patient is discussed with the doctor [22]. On the other hand, it is precisely this third response that leads us to formulate a question yet to be answered: can proactivity be encouraged during consultations? If so, could it result in reduced anxiety levels before US-guided procedures?

We verified the absence of significant associations between STAI-Trait scores, the studied demographic parameters, and Internet use. This finding was expected since the STAI-Trait scale reflects a more chronic predisposition to anxiety. A meta-analysis published by Schneider et al. [23] reviewed all published articles and unpublished dissertations between 1980 and 2005 that utilized STAI to address psychosocial interventions for cancer patients. The results for changes in STAI-Trait scores were equivocal, suggesting that pre-intervention stress is more accurately characterized by STAI-State. Thus, considering that the focus of our study was anxiety levels experienced moments before an invasive procedure, the STAI-State was expected to reflect the patients' anxiety levels more accurately.

It should also be noted that there is a high prevalence of psychiatric disorders such as depression and anxiety amongst cancer patients [24]. Despite the large knowledge gap in the medical literature and the absence of specific guidelines on the subject [25], health professionals involved in care are required to understand how to approach said conditions. While our study advances understanding of image-guided diagnostic procedures, it is important to note cancer patients are individuals with a significant anxiety burden that requires further investigation on the subject.

There are several other limitations in the present study that must be considered. Firstly, regarding the quality of communication by the referring physician, we draw attention to the lack of a structured diagnostic clinical interview. Additionally, we must consider the heterogeneity between the comparison groups, namely the different types of procedures in different organs and tissues. Finally, study participants were not directly asked whether they suffered from a psychiatric illness. While the authors found no evidence of consultations with clinical psychologists or psychiatrists in any of the patients' electronic medical records, participants were not directly asked about symptoms or previous traumatic events.

## Conclusion

Despite limitations, this study shows that higher anxiety scores before invasive US-guided procedures were found in female patients, in those who reported having had received insufficient information from their referring physician, and in those who considered online information to be unreliable or difficult to access. The recognition of this profile can guide measures to reduce anxiety in patients who will undergo an US-guided invasive procedure, by improving communication between patient and physician and providing accurate and easily accessible information online.

## Supporting information

**S1 Appendix.**
(DOCX)

**S1 Raw data.**
(XLS)

## Acknowledgments

The authors would like to thank the entire Nursing team that made this study possible.

## Author Contributions

**Conceptualization:** Marcio Meira, Almir Galvão Vieira Bitencourt, Demian Jungklaus Travesso, Paula Nicole Vieira Pinto Barbosa.

**Data curation:** Marcio Meira, Paula Nicole Vieira Pinto Barbosa.

**Formal analysis:** Marcio Meira, Almir Galvão Vieira Bitencourt, Paula Nicole Vieira Pinto Barbosa.

**Funding acquisition:** Marcio Meira.

**Investigation:** Marcio Meira.

**Methodology:** Marcio Meira, Almir Galvão Vieira Bitencourt.

**Project administration:** Marcio Meira, Demian Jungklaus Travesso, Paula Nicole Vieira Pinto Barbosa.

**Resources:** Marcio Meira.

**Supervision:** Almir Galvão Vieira Bitencourt, Rubens Chojniak.

**Validation:** Marcio Meira.

**Visualization:** Marcio Meira.

**Writing – original draft:** Marcio Meira.

**Writing – review & editing:** Marcio Meira, Almir Galvão Vieira Bitencourt, Demian Jungklaus Travesso, Rubens Chojniak, Paula Nicole Vieira Pinto Barbosa.

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
