## [Decision Letter · Decision Letter 0]

20 Oct 2021

PONE-D-21-27163Anxiety and internet research before percutaneous ultrasound-guided diagnostic proceduresPLOS ONE

Dear Dr. Meira,

Thank you for submitting your manuscript to PLOS ONE. After careful consideration, we feel that it has merit but does not fully meet PLOS ONE’s publication criteria as it currently stands. Therefore, we invite you to submit a revised version of the manuscript that addresses the points raised during the review process.

We look forward to receiving your revised manuscript.

Kind regards,

Marco Cascella

Academic Editor

PLOS ONE

Additional Editor Comments (if provided):

The article is well written but needs improvement. The reviewers did an excellent job and pointed out the various methodological gaps. Our goal is to give a high quality product to the press.

Authors should make some changes to improve the quality of the data presented. In particular, they must revise the introduction by inserting one or more sentences to bring the average reader closer to the issue. Moreover, they should offer a more articulate discussion of the results and perform a thorough revision of the language. Finally, the text must be revised according to the reviewers' suggestions (it is mandatory). The references suggested by one of the reviewers are excessive. Given the comments I got from the reviewers, my rating is major revision.

Journal Requirements:

2. Please provide additional details regarding participant consent. In the ethics statement in the Methods and online submission information, please ensure that you have specified what type you obtained (for instance, written or verbal, and if verbal, how it was documented and witnessed). If your study included minors, state whether you obtained consent from parents or guardians. If the need for consent was waived by the ethics committee, please include this information

Reviewers' comments:

Reviewer's Responses to Questions

**Comments to the Author**

1. Is the manuscript technically sound, and do the data support the conclusions?

Reviewer #1: Yes

Reviewer #2: Yes

Reviewer #3: Partly

Reviewer #4: Yes

2. Has the statistical analysis been performed appropriately and rigorously? 

Reviewer #1: Yes

Reviewer #2: Yes

Reviewer #3: Yes

Reviewer #4: I Don't Know

3. Have the authors made all data underlying the findings in their manuscript fully available?

Reviewer #1: Yes

Reviewer #2: Yes

Reviewer #3: Yes

Reviewer #4: Yes

4. Is the manuscript presented in an intelligible fashion and written in standard English?

Reviewer #1: No

Reviewer #2: Yes

Reviewer #3: Yes

Reviewer #4: Yes

5. Review Comments to the Author

Reviewer #1: I am glad to review and assess this exciting article, entitled, “Anxiety and internet research before percutaneous ultrasound-guided diagnostic procedures”. Invasive procedures guided by ultrasound (US) are part of the routine medical diagnostic investigation. The lack of knowledge related to technical aspects about them can lead the patient to seek complementary information on the internet, which can trigger anxiety. However, the intersection between the areas of Radiology and Psychology is poorly studied. Here we show the profile of an anxious patient before an US-guided intervention.

The organization of this article is good and partly satisfactory. The Introduction section, methodology portions are adequate. I suggest the authors improve these parts overall to enhance the work quality. As suggested, I recommend that authors do a little more work and add the latest literature to support the study. I accept this manuscript after minor revision, as I have recommended.

Some valuable comments are given below;

• The manuscript needs language, grammar, and syntactic editing. The English language usage should be checked by a fluent English speaker (Writing quality is inferior. Numerous grammatical mistakes and meaningless sentences)

• The primary defect of this study is the debate or argument is not clearly stated in the introduction session. Hence, the contribution is weak in this manuscript. I would suggest the author enhance your theoretical discussion and your debate or argument.

• The research gap is not evident and appropriate.

• Must add much more explanations and interpretations for the results, which are not enough

• It is suggested to compare the results of the present research with some similar studies which is done before (more justification is needed)

• Should add a flowchart to the article to show the research methodology

Reviewer #2: This paper is well written and enriches the existing literature. The methodology is sound and has been applied before. The results are presented well and discussed. Therefore, I will recommend the acceptance of the paper.

Reviewer #3: I have evaluated this interesting study entitled, "Anxiety and internet research before percutaneous ultrasound-guided diagnostic procedures."

is an informative research study. I suggest some minor corrections to check the typo errors in writing to enhance the English quality to reach the scientific merit for the publication of this study.

This article describes that invasive procedures guided by ultrasound (US) are part of the routine medical diagnostic investigation. The lack of knowledge related to technical aspects about them can lead the patient to seek complementary information on the internet, which can trigger anxiety. However, the intersection between the areas of Radiology and Psychology is poorly studied. Here we show the profile of an anxious patient before an US-guided intervention.

I am in favor of this study and will recommend for publication. However, the authors need to revise the manuscript and work according to my suggestions to enhance the quality. I will accept this paper for publication after these minor changes as suggested below.

Introduction and literation sections

I recommend the authors add suggested articles in the introduction and literature sections. These research articles have identified health-related topics I believe it will improve the quality of your work. I strongly suggested them to improve this section a bit more. I advise authors to revisit their introduction and literature sections of the recommended studies and cite these studies to enhance your research study's quality to reach scientific merit for publication.

Wang, C., Wang, D., Duan, K., & Mubeen, R. (2021). Global financial crisis, smart lockdown strategies, and the COVID-19 spillover impacts: A global perspective implications from Southeast Asia. Front Psychiatry, 12, 1-14. doi:10.3389/fpsyt.2021.643783

Abbas, J., Raza, S., Nurunnabi, M., Minai, M. S., & Bano, S. (2019). The Impact of Entrepreneurial Business Networks on Firms’ Performance Through a Mediating Role of Dynamic Capabilities. Sustainability, 11(11), 3006. doi:10.3390/su11113006

NeJhaddadgar, N., Ziapour, A., Zakkipour, G., Abolfathi, M., & Shabani, M. (2020, Nov 13). Effectiveness of telephone-based screening and triage during COVID-19 outbreak in the promoted primary healthcare system: a case study in Ardabil province, Iran. Z Gesundh Wiss, 1-6. https://doi.org/10.1007/s10389-020-01407-8

Abbas, J., Aqeel, M., Abbas, J., Shaher, B., A, J., Sundas, J., & Zhang, W. (2019, Feb 1). The moderating role of social support for marital adjustment, depression, anxiety, and stress: Evidence from Pakistani working and nonworking women. J Affect Disord, 244, 231-238. https://doi.org/10.1016/j.jad.2018.07.071

Literature

I want to see publish this creative study after some corrections. I have endorsed this study as; it deserves the merit for publication. However, I suggest the authors make minor corrections according to my advice. Please read the suggested studies and cite them in the introduction, literature, and method sections. How corporate social responsibility, innovation and social media and internet use is helpful. Add few lines in the introduction and literature sections. How companies are practicing CSR, business, entrepreneurial networks with innovation and knowledge sharing to improve the business performance and provide better healthcare medicines?

Azizi, M. R., Atlasi, R., Ziapour, & Naemi, R. (2021). Innovative human resource management strategies during the COVID-19 pandemic: A systematic narrative review approach. Heliyon, 7(6), e07233. doi:10.1016/j.heliyon.2021.e07233

Abbas, J., Zhang, Q., Hussain, I., Akram, S., Afaq, A., & Shad, M. A. (2020). Sustainable Innovation in Small Medium Enterprises: The Impact of Knowledge Management on Organizational Innovation through a Mediation Analysis by Using SEM Approach. Sustainability, 12(6), 2407. doi:https://doi.org/10.3390/su12062407

Azadi, N. A., Ziapour, A., Lebni, J. Y., Irandoost, S. F., & Chaboksavar, F. (2021). The effect of education based on health belief model on promoting preventive behaviors of hypertensive disease in staff of the Iran University of Medical Sciences. Archives of Public Health, 79(1), 69. doi:10.1186/s13690-021-00594-4

Abbas, J., Hussain, I., Hussain, S., Akram, S., Shaheen, I., & Niu, B. (2019). The Impact of Knowledge Sharing and Innovation upon Sustainable Performance in Islamic Banks: A Mediation Analysis through an SEM Approach. Sustainability, 11(15), 4049. doi:10.3390/su11154049

Materials and Methods

The results section of the paper presents a good view of the study. This work presents a notable investigation on a selected topic. I suggest the authors to present high quality graphs. By including some graphical presentations will improve the quality of this study. Please see the proposed studies and see the graphical representation. Improve your work like these studies and cite them in this section.

Paulson, K. R., Kamath, A. M., Alam, T., Bienhoff, K., Abady, G. G., . . . Kassebaum, N. J. (2021). Global, regional, and national progress towards Sustainable Development Goal 3.2 for neonatal and child health: all-cause and cause-specific mortality findings from the Global Burden of Disease Study 2019. The Lancet, 1-36. doi:10.1016/s0140-6736(21)01207-1

Abbas, J., Aqeel, M., Ling, J., Ziapour, A., Raza, M. A., & Rehna, T. (2020). Exploring the relationship between intimate partner abuses, resilience, psychological, and physical health problems in Pakistani married couples: a perspective from the collectivistic culture. Sexual and Relationship Therapy, 35, 1-30. https://doi.org/10.1080/14681994.2020.1851673

Abbas, J., Aman, J., Nurunnabi, M., & Bano, S. (2019). The Impact of Social Media on Learning Behavior for Sustainable Education: Evidence of Students from Selected Universities in Pakistan. Sustainability, 11(6). https://doi.org/10.3390/su11061683

Abbas, J., Aqeel, M., Jaffar, A., Nurunnabi, M., & Bano, S. (2019, 2019/07/01). Tinnitus perception mediates the relationship between physiological and psychological problems among patients. Journal of Experimental Psychopathology, 10(3), 2043808719858559. https://doi.org/10.1177/2043808719858559

Discussion

I suggest the authors to discuss the effects of the COVID-19. I suggest you to cite these studies. Read the proposed studies to improve your results and discussion section. See the recommended studies and improve your sections.

Su, Z., McDonnell, D., Wen, J., Kozak, M., Šegalo, S., . . . Xiang, Y.-T. (2021). Mental health consequences of COVID-19 media coverage: the need for effective crisis communication practices. Globalization and Health, 17(1), 4. doi:10.1186/s12992-020-00654-4

Aqeel, M., Shuja, K. H., Rehna, T., Ziapour, A., Yousaf, I., & Karamat, T. (2021). The Influence of Illness Perception, Anxiety and Depression Disorders on Students Mental Health during COVID-19 Outbreak in Pakistan: A Web-Based Cross-Sectional Survey. International Journal of Human Rights in Healthcare, 14, 1-14.

Abbas, J. (2020). The Impact of Coronavirus (SARS-CoV2) Epidemic on Individuals Mental Health: The Protective Measures of Pakistan in Managing and Sustaining Transmissible Disease. Psychiatr Danub, 32(3-4), 472-477. https://doi.org/10.24869/psyd.2020.472

Conclusion

I suggest you make a separate heading of the conclusion and do not mix it with implications.

Policy Recommendations

I again recommend you to make a separate heading of the Policy Recommendations.

The conclusion section is acceptable. Overall, this presents a good piece of research work. I recommend that authors do a little more work and revise this article accordingly. I suggest the authors check English quality and fix some weak sentences. If you have already taken English editing service, ask them to recheck the quality to meet scientific merit for publication. I endorse this manuscript for publication after minor corrections, as suggested.

Reviewer #4: Thank you for submitting the manuscript. I have read your manuscript very carefully. The theme you are addressing is an important one that investigates the doctor-patient relationship. I am convinced that small adjustments can increase the quality and readability of the paper.First of all I would like you to enter the protocol number of the approval by the Ethics Committee of the study.I would also like to have more information about the patient's medical history. I'll explain. The paper does not talk about the psychological history of patients, but only about their self-definition as pro-active or not.This definition is too reductive to understand the psychological situation of the patients involved. The psychological history of the subjects would provide us with important information to better understand the possible origin of anxiety and possible confounding factors.Have any traumatic factors been investigated in the history of the subjects (war veterans, child abuse, road accidents ...)? Have you asked about the possible use of anxiolytic drugs? Have you asked if the subjects have ever been in psychotherapy?If you have this information, in my opinion, it must be explained in the paper and it must be correlated with the results. If, on the other hand, you do not have one, it is necessary to insert this gap among the limitations of the study.I am sure that with these small suggestions the paper can be improved. I hope these comments are helpful to you.

Kind Regards

6. PLOS authors have the option to publish the peer review history of their article (what does this mean?). If published, this will include your full peer review and any attached files.

Reviewer #1: **Yes: **KASHIF ABBASS

Reviewer #2: No

Reviewer #3: No

Reviewer #4: No

---

## [Author Response · Author response to Decision Letter 0]

4 Dec 2021

Reviewer #1

1. The manuscript needs language, grammar, and syntactic editing. The English language usage should be checked by a fluent English speaker (Writing quality is inferior. Numerous grammatical mistakes and meaningless sentences)

The authors agree with the statements. The entire article was submitted for review by a native English speaker.

2. The primary defect of this study is the debate or argument is not clearly stated in the introduction session. Hence, the contribution is weak in this manuscript. I would suggest the author enhance your theoretical discussion and your debate or argument. The research gap is not evident and appropriate.

The authors agree. To clarify the central theme of the study, the following excerpt was added to the first paragraph of the introduction: “Although image-guided procedures and the use of the Internet as a resource are ingrained in the modern world, few studies have correlated the two with the presence of anxiety before invasive procedures.”

3. Must add much more explanations and interpretations for the results, which are not enough. It is suggested to compare the results of the present research with some similar studies which is done before (more justification is needed)

The authors agree. More time was spent on improving the discussion section to bring more clarity when connecting our study with previous publications.

• Should add a flowchart to the article to show the research methodology

The authors agree and the text was edited to include the flowchart represented in Figure 2.

Reviewer #2

This paper is well written and enriches the existing literature. The methodology is sound and has been applied before. The results are presented well and discussed. Therefore, I will recommend the acceptance of the paper.

The authors are grateful for the comments.

Reviewer #3: 

I have evaluated this interesting study entitled, "Anxiety and internet research before percutaneous ultrasound-guided diagnostic procedures." is an informative research study. I suggest some minor corrections to check the typo errors in writing to enhance the English quality to reach the scientific merit for the publication of this study.

This article describes that invasive procedures guided by ultrasound (US) are part of the routine medical diagnostic investigation. The lack of knowledge related to technical aspects about them can lead the patient to seek complementary information on the internet, which can trigger anxiety. However, the intersection between the areas of Radiology and Psychology is poorly studied. Here we show the profile of an anxious patient before an US-guided intervention.

I am in favor of this study and will recommend for publication. However, the authors need to revise the manuscript and work according to my suggestions to enhance the quality. I will accept this paper for publication after these minor changes as suggested below.

Introduction and literation sections

I recommend the authors add suggested articles in the introduction and literature sections. These research articles have identified health-related topics I believe it will improve the quality of your work. I strongly suggested them to improve this section a bit more. I advise authors to revisit their introduction and literature sections of the recommended studies and cite these studies to enhance your research study's quality to reach scientific merit for publication.

Wang, C., Wang, D., Duan, K., & Mubeen, R. (2021). Global financial crisis, smart lockdown strategies, and the COVID-19 spillover impacts: A global perspective implications from Southeast Asia. Front Psychiatry, 12, 1-14. doi:10.3389/fpsyt.2021.643783

Abbas, J., Raza, S., Nurunnabi, M., Minai, M. S., & Bano, S. (2019). The Impact of Entrepreneurial Business Networks on Firms’ Performance Through a Mediating Role of Dynamic Capabilities. Sustainability, 11(11), 3006. doi:10.3390/su11113006

NeJhaddadgar, N., Ziapour, A., Zakkipour, G., Abolfathi, M., & Shabani, M. (2020, Nov 13). Effectiveness of telephone-based screening and triage during COVID-19 outbreak in the promoted primary healthcare system: a case study in Ardabil province, Iran. Z Gesundh Wiss, 1-6. https://doi.org/10.1007/s10389-020-01407-8

Abbas, J., Aqeel, M., Abbas, J., Shaher, B., A, J., Sundas, J., & Zhang, W. (2019, Feb 1). The moderating role of social support for marital adjustment, depression, anxiety, and stress: Evidence from Pakistani working and nonworking women. J Affect Disord, 244, 231-238. https://doi.org/10.1016/j.jad.2018.07.071

Literature

I want to see publish this creative study after some corrections. I have endorsed this study as; it deserves the merit for publication. However, I suggest the authors make minor corrections according to my advice. Please read the suggested studies and cite them in the introduction, literature, and method sections. How corporate social responsibility, innovation and social media and internet use is helpful. Add few lines in the introduction and literature sections. How companies are practicing CSR, business, entrepreneurial networks with innovation and knowledge sharing to improve the business performance and provide better healthcare medicines?

Azizi, M. R., Atlasi, R., Ziapour, & Naemi, R. (2021). Innovative human resource management strategies during the COVID-19 pandemic: A systematic narrative review approach. Heliyon, 7(6), e07233. doi:10.1016/j.heliyon.2021.e07233

Abbas, J., Zhang, Q., Hussain, I., Akram, S., Afaq, A., & Shad, M. A. (2020). Sustainable Innovation in Small Medium Enterprises: The Impact of Knowledge Management on Organizational Innovation through a Mediation Analysis by Using SEM Approach. Sustainability, 12(6), 2407. doi:https://doi.org/10.3390/su12062407

Azadi, N. A., Ziapour, A., Lebni, J. Y., Irandoost, S. F., & Chaboksavar, F. (2021). The effect of education based on health belief model on promoting preventive behaviors of hypertensive disease in staff of the Iran University of Medical Sciences. Archives of Public Health, 79(1), 69. doi:10.1186/s13690-021-00594-4

Abbas, J., Hussain, I., Hussain, S., Akram, S., Shaheen, I., & Niu, B. (2019). The Impact of Knowledge Sharing and Innovation upon Sustainable Performance in Islamic Banks: A Mediation Analysis through an SEM Approach. Sustainability, 11(15), 4049. doi:10.3390/su11154049

Materials and Methods

The results section of the paper presents a good view of the study. This work presents a notable investigation on a selected topic. I suggest the authors to present high quality graphs. By including some graphical presentations will improve the quality of this study. Please see the proposed studies and see the graphical representation. Improve your work like these studies and cite them in this section.

Paulson, K. R., Kamath, A. M., Alam, T., Bienhoff, K., Abady, G. G., . . . Kassebaum, N. J. (2021). Global, regional, and national progress towards Sustainable Development Goal 3.2 for neonatal and child health: all-cause and cause-specific mortality findings from the Global Burden of Disease Study 2019. The Lancet, 1-36. doi:10.1016/s0140-6736(21)01207-1

Abbas, J., Aqeel, M., Ling, J., Ziapour, A., Raza, M. A., & Rehna, T. (2020). Exploring the relationship between intimate partner abuses, resilience, psychological, and physical health problems in Pakistani married couples: a perspective from the collectivistic culture. Sexual and Relationship Therapy, 35, 1-30. https://doi.org/10.1080/14681994.2020.1851673

Abbas, J., Aman, J., Nurunnabi, M., & Bano, S. (2019). The Impact of Social Media on Learning Behavior for Sustainable Education: Evidence of Students from Selected Universities in Pakistan. Sustainability, 11(6). https://doi.org/10.3390/su11061683

Abbas, J., Aqeel, M., Jaffar, A., Nurunnabi, M., & Bano, S. (2019, 2019/07/01). Tinnitus perception mediates the relationship between physiological and psychological problems among patients. Journal of Experimental Psychopathology, 10(3), 2043808719858559. https://doi.org/10.1177/2043808719858559

The authors agree and the text was edited to include the flowchart represented in Figure 2.

Discussion

I suggest the authors to discuss the effects of the COVID-19. I suggest you to cite these studies. Read the proposed studies to improve your results and discussion section. See the recommended studies and improve your sections.

Su, Z., McDonnell, D., Wen, J., Kozak, M., Šegalo, S., . . . Xiang, Y.-T. (2021). Mental health consequences of COVID-19 media coverage: the need for effective crisis communication practices. Globalization and Health, 17(1), 4. doi:10.1186/s12992-020-00654-4

Aqeel, M., Shuja, K. H., Rehna, T., Ziapour, A., Yousaf, I., & Karamat, T. (2021). The Influence of Illness Perception, Anxiety and Depression Disorders on Students Mental Health during COVID-19 Outbreak in Pakistan: A Web-Based Cross-Sectional Survey. International Journal of Human Rights in Healthcare, 14, 1-14.

Abbas, J. (2020). The Impact of Coronavirus (SARS-CoV2) Epidemic on Individuals Mental Health: The Protective Measures of Pakistan in Managing and Sustaining Transmissible Disease. Psychiatr Danub, 32(3-4), 472-477. https://doi.org/10.24869/psyd.2020.472

The authors are grateful for the constructive feedback and have read all the above studies. After careful consideration we selected the following articles to include in our study. The first one was entitled “Abbas, J., Aqeel, M., Abbas, J., Shaher, B., A, J., Sundas, J., & Zhang, W. (2019, Feb 1). The moderating role of social support for marital adjustment, depression, anxiety, and stress: Evidence from Pakistani working and nonworking women. J Affect Disord, 244, 231-238. https://doi.org/10.1016/j.jad.2018.07.071”. It fits perfectly with our argument that correlates higher anxiety in the female gender, as it follows: “Preoperatively, multiple parameters were evaluated including: patients’ age, sex gender, occupation, education level, type of surgery, and whether they patient had previously undergone a similar procedure. Respondents completed the STAI questionnaire for anxiety psychometry in the waiting room right before their intervention. Of all the parameters examined, only the female gender was found to be positively correlated with anxiety levels. This finding has also been reported in other parts of the world [10]. Jafar et al. used STAI to evaluate 300 pre-surgical patients in Pakistan and found higher levels of anxiety in females. However, considering anxiety questionnaires are self-administered, female patients may be more inclined to admit to having anxiety than males patients. It is therefore necessary to exercise caution before stating that these data represent an innate difference in anxiety levels between genders [11].”

Conclusion

I suggest you make a separate heading of the conclusion and do not mix it with implications.

Policy Recommendations

I again recommend you to make a separate heading of the Policy Recommendations.

Although the author agrees with the comment and think it would be more coherent to separate the policy recommendations and implications from the conclusion, this paper follows the recommendations and structure set out in PLOSONE guideline.

The conclusion section is acceptable. Overall, this presents a good piece of research work. I recommend that authors do a little more work and revise this article accordingly. I suggest the authors check English quality and fix some weak sentences. If you have already taken English editing service, ask them to recheck the quality to meet scientific merit for publication. I endorse this manuscript for publication after minor corrections, as suggested.

The authors agree. To clarify the central theme of the study, the following excerpt was added to the first paragraph of the introduction: “Although image-guided procedures and the use of the Internet as a resource are ingrained in the modern world, few studies have correlated the two with the presence of anxiety before invasive procedures.”. Additionally, the entire article was submitted for review by a native English speaker.

Reviewer #4

Thank you for submitting the manuscript. I have read your manuscript very carefully. The theme you are addressing is an important one that investigates the doctor-patient relationship. I am convinced that small adjustments can increase the quality and readability of the paper. 

1. First of all I would like you to enter the protocol number of the approval by the Ethics Committee of the study. 

The authors agree and added the following text: “(certificate number approval by the ethics committee 66022517.8.0000.5432, appraisal 2.063.731).” - Study Design.

2. I would also like to have more information about the patient's medical history. I'll explain. The paper does not talk about the psychological history of patients, but only about their self-definition as pro-active or not. This definition is too reductive to understand the psychological situation of the patients involved. The psychological history of the subjects would provide us with important information to better understand the possible origin of anxiety and possible confounding factors. Have any traumatic factors been investigated in the history of the subjects (war veterans, child abuse, road accidents ...)? Have you asked about the possible use of anxiolytic drugs? Have you asked if the subjects have ever been in psychotherapy? If you have this information, in my opinion, it must be explained in the paper and it must be correlated with the results. If, on the other hand, you do not have one, it is necessary to insert this gap among the limitations of the study. I am sure that with these small suggestions the paper can be improved. I hope these comments are helpful to you.

This observation is very important and the authors are grateful. During the study, we were careful to look for appointments with psychologists or psychiatrists by the participants. Despite not being an inclusion/exclusion criterion, we found that no patient had had an appointment with these professionals at the hospital where the invasive procedure was performed. This does not prevent an important limitation of our study since aspects of psychiatric illnesses were not directly addressed or questioned. These points have been added to the Discussion session, in limitations, as follows: “Finally, there were no direct questions included regarding the presence of psychiatric illness among study participants. While the authors found no evidence of consultations with clinical psychologists or psychiatrists in the patients’ electronic medical records, participants were not directly asked about symptoms or previous traumatic events.”

---

## [Editor Report · Decision Letter 1]

7 Jan 2022

PONE-D-21-27163R1Anxiety and internet research before percutaneous ultrasound-guided diagnostic proceduresPLOS ONE

Dear Dr. Meira,

Thank you for submitting your manuscript to PLOS ONE. After careful consideration, we feel that it has merit but does not fully meet PLOS ONE’s publication criteria as it currently stands. Therefore, we invite you to submit a revised version of the manuscript that addresses the points raised during the review process.

We look forward to receiving your revised manuscript.

Kind regards,

Marco Cascella

Academic Editor

PLOS ONE

Additional Editor Comments:

I suggest further changes

1. TITLE. Include the type of the study and setting.. For example: "A prospective cohort study ..."

2. ABSTRACT. Include p-values

3. LIMITATIONS (It deserves a dedicated paragraph). Emphasizes that cancer patients are individuals with a significant anxiety burden.

3. CONCLUSION. Add "Despite limitations, this study shows that ..."

4. REFERENCES. Many references are more than 20 years old. This is a gap for the article. Add, other more recent references:

a. about anxiety: doi: 10.1097/RMR.0000000000000238; doi 10.18632/oncotarget.17238; doi: 10.1002/da.23115

b. about internet-based patients' searching: doi: 10.1176/appi.ps.201800495; doi: 10.1371/journal.pone.0261471.

Finally, I suggest another round of lexical correction. The style still doesn't sound.
---

## [Author Response · Author response to Decision Letter 1]

12 Feb 2022

Editor Comments

I suggest further changes

1. TITLE. Include the type of the study and setting.. For example: "A prospective cohort study ..."

The study title has been changed to: “Relationship Between Anxiety and Internet Searches Before Percutaneous Ultrasound-guided Diagnostic Procedures: A Prospective Cohort Study”

2. ABSTRACT. Include p-values

P-values were included.

3. LIMITATIONS (It deserves a dedicated paragraph). Emphasizes that cancer patients are individuals with a significant anxiety burden.

Prior to specific limitations session, the following was added: “It should also be noted that there is a high prevalence of psychiatric disorders such as depression and anxiety amongst cancer patients [22]. This universe requires understanding from health professionals involved in care, despite the large knowledge gap in the medical literature and the absence of specific guidelines on the subject [23]. While our study advances understanding of image-guided diagnostic procedures, it is important to note cancer patients are individuals with a significant anxiety burden that requires further investigations on the subject.

3. CONCLUSION. Add "Despite limitations, this study shows that ..."

Added phrase as follows: “Despite limitations, this study shows that higher anxiety scores before invasive US-guided procedures were found in female patients, in those who reported having had received insufficient information from their referring physician, and in those who considered online information to be unreliable or difficult to access. The recognition of this profile can guide measures to reduce anxiety in patients who will undergo an US-guided invasive procedure, by improving patient-physician communication and providing accurate and easily accessible information online.”

4. REFERENCES. Many references are more than 20 years old. This is a gap for the article. Add, other more recent references: 

a. about anxiety: doi: 10.1097/RMR.0000000000000238; doi 10.18632/oncotarget.17238; doi: 10.1002/da.23115 b. about internet-based patients' searching: doi: 10.1176/appi.ps.201800495; doi: 10.1371/journal.pone.0261471.

The authors agree with this observation. The reference “Bull Med Libr Assoc. 2001 Oct;89(4):397-9. PMID: 11837263; PMCID: PMC57970” has been replaced by “Top Magn Reson Imaging. 2020 Aug;29(4):197-201. doi: 10.1097/RMR.0000000000000238. PMID: 32472820”. Another updated reference was added (Psychiatr Serv. 2019 Apr 1;70(4):324-328.”).

We are especially grateful for the suggested article " PLoS One. 2021 Dec 31;16(12):e0261471" as it fits properly into the following manuscript excerpt: “Three potential responses to patients who utilize online searches prior to consultation were discussed: 1) the healthcare professional feels threatened by the information and responds by defensively stating his "expert opinion"; 2) the healthcare professional and the patient cooperate in the analysis of online information; 3) the healthcare professional guides patients to reliable sites on the Internet. Previous studies suggest positive results when the information brought from the Internet by the patient is discussed together with the doctor [21]. On the other hand, it is precisely in this third response that our study supports and leads us to formulate a question yet to be answered: can proactivity be encouraged during consultations? If so, could it result in reduced anxiety before US-guided procedures?”

Finally, I suggest another round of lexical correction. The style still doesn't sound.

A new spelling and writing style review was requested from a professional who is native and fluent in the English language.

---

## [Decision Letter · Decision Letter 2]

7 Jun 2022

PONE-D-21-27163R2Relationship Between Anxiety and Internet Searches Before Percutaneous Ultrasound-guided Diagnostic Procedures: A Prospective Cohort StudyPLOS ONE

Dear Dr. Meira,

Thank you for submitting your manuscript to PLOS ONE. After careful consideration, we feel that it has merit but does not fully meet PLOS ONE’s publication criteria as it currently stands. Therefore, we invite you to submit a revised version of the manuscript that addresses the points raised during the review process.

We look forward to receiving your revised manuscript.

Kind regards,

Francisco Sampaio, Ph.D.

Guest Editor

PLOS ONE

Additional Editor Comments (if provided):

Dear authors,

After being sent to three independent reviewers, they considered the paper would need a major revision in order to potentially be published.

Thus, please, read carefully the comments of the reviewers and try to give response to them.

Best regards,

Francisco Sampaio

Reviewers' comments:

Reviewer's Responses to Questions

**Comments to the Author**

1. If the authors have adequately addressed your comments raised in a previous round of review and you feel that this manuscript is now acceptable for publication, you may indicate that here to bypass the “Comments to the Author” section, enter your conflict of interest statement in the “Confidential to Editor” section, and submit your "Accept" recommendation.

Reviewer #5: All comments have been addressed

Reviewer #6: (No Response)

Reviewer #7: (No Response)

2. Is the manuscript technically sound, and do the data support the conclusions?

Reviewer #5: Yes

Reviewer #6: Partly

Reviewer #7: Yes

3. Has the statistical analysis been performed appropriately and rigorously? 

Reviewer #5: No

Reviewer #6: I Don't Know

Reviewer #7: I Don't Know

4. Have the authors made all data underlying the findings in their manuscript fully available?

Reviewer #5: Yes

Reviewer #6: Yes

Reviewer #7: No

5. Is the manuscript presented in an intelligible fashion and written in standard English?

Reviewer #5: Yes

Reviewer #6: No

Reviewer #7: Yes

6. Review Comments to the Author

Reviewer #5: * It is a very interesting and very current topic. Health systems must find strategies to increase the well-being of their patients. This work can help to find these strategies, after identifying the causes.

* Despite not being native to the English language, it seems to me that English should be revised.

* On page 12, section "Statistical Methods", authors should add, right after the phrase "The non-parametric Mann-Whitney

test was used to compare two subgroups, and the Kruskal-Wallis test was used when three subgroups were analyzed.", the following:

"In the 3 group comparison, whenever statistically significant differences were detected, the Kruskal-Wallis multiple comparison test was used."

Authors should not use the Mann-Whitney test to compare groups 2 to 2, as this increases the probability of type I error from 0.05 to 0.143 (as it is related to the number of comparisons to be performed). They should use the Kruskal-Wallis multiple comparison tests, already available in SPSS version 20.0.

* The titles of tables 2, 3, 4 and 5 are not correct. In any of the tables, what the authors are doing is comparing anxiety levels (STAI-State and STAI-Trait) between groups. They are not studying the correlation.

Reviewer #6: It seems, the manuscript do not follow scientific writing for an article such as topic presentations and study design and figure 1, raw presentation.

Reviewer #7: It is a current exciting topic and manuscript. While not the most original work, it adds exciting data concerning patient-centered care during diagnostic procedures. It provides essential information to oncology specialty professionals and general practice. However, the manuscript presents some relevant weaknesses.

A linguistic revision is suggested to make the reading more accessible and precise.

Abstract: Reduce background (the last two sentences are not needed) and improve methods, results, and conclusion

Introduction: The introduction should be improved with more references related to diagnostic medical procedures and online search anxiety. There are scientific papers available.

Methods: Please organize better methods section regarding Sample or participants, measures (information regarding STAI is repetitive), proceedings (including dates for data collecting), ethical considerations, and Statistical analyses.

Results: Did the authors considered performed Hierarchical linear regression analyses to explore potential predictors of patients’ anxiety before US-guided invasive procedures?

Considering STAI_T, authors could also use descriptive data of patients with clinically relevant anxiety and without clinically relevant anxiety.

Discussion: The discussion tries to be exhaustive, with references to other studies. However, the complete information related to cancer is not clear since the current sample did not have a cancer diagnosis. This need to be clarified perhaps in the methods section

making it very long and with some unnecessary indications (for example, values of the statistics of the studies with which it compares its results). The discussion should also include limitations and indications for future studies.

A linguistic revision is suggested to make the reading more accessible and precise.

7. PLOS authors have the option to publish the peer review history of their article (what does this mean?). If published, this will include your full peer review and any attached files.

Reviewer #5: No

Reviewer #6: **Yes: **Ali Bikmoradi

Reviewer #7: No

---

## [Author Response · Author response to Decision Letter 2]

22 Jul 2022

Reviewer #5

* It is a very interesting and very current topic. Health systems must find strategies to increase the well-being of their patients. This work can help to find these strategies, after identifying the causes.

1. *Despite not being native to the English language, it seems to me that English should be revised.

The authors agree with the statements. The entire article was submitted for review by a native English speaker.

2. *On page 12, section "Statistical Methods", authors should add, right after the phrase "The non-parametric Mann-Whitney test was used to compare two subgroups, and the Kruskal-Wallis test was used when three subgroups were analyzed.", the following:

"In the 3 group comparison, whenever statistically significant differences were detected, the Kruskal-Wallis multiple comparison test was used."

Authors should not use the Mann-Whitney test to compare groups 2 to 2, as this increases the probability of type I error from 0.05 to 0.143 (as it is related to the number of comparisons to be performed). They should use the Kruskal-Wallis multiple comparison tests, already available in SPSS version 20.0.

The authors agree with the statements. The suggested sentence was included in its entirety in the manuscript.

3. * The titles of tables 2, 3, 4 and 5 are not correct. In any of the tables, what the authors are doing is comparing anxiety levels (STAI-State and STAI-Trait) between groups. They are not studying the correlation.

The authors agree with the indicated inadequacy. The term relationship has been changed to correlation.

Reviewer #6

It seems, the manuscript do not follow scientific writing for an article such as topic presentations and study design and figure 1, raw presentation.

The authors appreciate the reviewer's comment. However, we did our best to follow scientific methodology, especially in writing. After going through corrections from 7 different reviewers and 2 editors, we believe we have done a good job. Thanks again for the comment.

Reviewer #7

It is a current exciting topic and manuscript. While not the most original work, it adds exciting data concerning patient-centered care during diagnostic procedures. It provides essential information to oncology specialty professionals and general practice. However, the manuscript presents some relevant weaknesses.

1. A linguistic revision is suggested to make the reading more accessible and precise.

The authors agree with the statements. The entire article was submitted for review by a native English speaker.

2. Abstract: Reduce background (the last two sentences are not needed) and improve methods, results, and conclusion

The authors are grateful for the observations made by the reviewer. The abstract was reduced, and the other sections were revised to clarify the ideas presented.

3. Introduction: The introduction should be improved with more references related to diagnostic medical procedures and online search anxiety. There are scientific papers available.

The authors agree with the observed fact and chose to add the following article to the manuscript: "Swire-Thompson B, Lazer D. Public Health and Online Misinformation: Challenges and Recommendations. Annu Rev Public Health. 2020 Apr 2;41:433-451 doi: 10.1146/annurev-publhealth-040119-094127. Epub 2019 Dec 24. PMID: 31874069". It is an excellent source that supports the misinterpretations that online information can lead to.

4. Methods: Please organize better methods section regarding Sample or participants, measures (information regarding STAI is repetitive), proceedings (including dates for data collecting), ethical considerations, and Statistical analyses.

The authors eliminated some repetitive information about the STAI questionnaire and also added the study data collection date to the manuscript in the Methods / Study Design section.

5. Results: Did the authors considered performed Hierarchical linear regression analyses to explore potential predictors of patients’ anxiety before US-guided invasive procedures?

Considering STAI_T, authors could also use descriptive data of patients with clinically relevant anxiety and without clinically relevant anxiety.

The authors are grateful for the questions raised. Hierarchical linear regression analysis was considered, however, as it is a topic such as anxiety, with countless variables, the authors opted for a statistical method rather than a model comparison. The STAI_T was used in our study basically because it was attached to the STAI_S in its original questionnaire. However, as the image-guided invasive procedure is a one-off event, its use could have been dispensed with. There were no significant correlations with this questionnaire. We chose to keep it in the manuscript precisely to raise this discussion.

6. Discussion: The discussion tries to be exhaustive, with references to other studies. However, the complete information related to cancer is not clear since the current sample did not have a cancer diagnosis. This need to be clarified perhaps in the methods section

making it very long and with some unnecessary indications (for example, values of the statistics of the studies with which it compares its results). The discussion should also include limitations and indications for future studies.

The authors are grateful for the comments raised by the reviewer. The sample of participants included cancer patients (45 patients, about 34% of the total), as shown in Table 1. There was no correlation with anxiety levels , as shown in the Results section: "There were no significant associations between STAI anxiety scores and age, education level, age, past experiences of health treatment of the medical status, or proposed procedure or subjective assessments of the regularity of care (P > 005)." As the reviewer correctly pointed out, this issue needs clarification, this point being addressed in the Discussion, in limitations, in the following excerpt: "It should also be noted that there is a high prevalence of psychiatric disorders such as depression and anxiety amongst cancer patients [23 ]. Despite the large knowledge gap in the medical literature and the absence of specific guidelines on the subject [24], health professionals in care are required to understand how to approach said conditions."

A linguistic revision is suggested to make the reading more accessible and precise.

The authors agree with the statements. The entire article was submitted for review by a native English speaker.

---

## [Decision Letter · Decision Letter 3]

9 Aug 2022

PONE-D-21-27163R3Relationship Between Anxiety and Internet Searches Before Percutaneous Ultrasound-guided Diagnostic Procedures: A Prospective Cohort StudyPLOS ONE

Dear Dr. Meira,

Thank you for submitting your manuscript to PLOS ONE. After careful consideration, we feel that it has merit but does not fully meet PLOS ONE’s publication criteria as it currently stands. Therefore, we invite you to submit a revised version of the manuscript that addresses the points raised during the review process.

We look forward to receiving your revised manuscript.

Kind regards,

Francisco Sampaio, Ph.D.

Guest Editor

PLOS ONE

Reviewers' comments:

Reviewer's Responses to Questions

**Comments to the Author**

1. If the authors have adequately addressed your comments raised in a previous round of review and you feel that this manuscript is now acceptable for publication, you may indicate that here to bypass the “Comments to the Author” section, enter your conflict of interest statement in the “Confidential to Editor” section, and submit your "Accept" recommendation.

Reviewer #5: (No Response)

Reviewer #6: All comments have been addressed

2. Is the manuscript technically sound, and do the data support the conclusions?

Reviewer #5: Yes

Reviewer #6: Partly

3. Has the statistical analysis been performed appropriately and rigorously? 

Reviewer #5: Yes

Reviewer #6: I Don't Know

4. Have the authors made all data underlying the findings in their manuscript fully available?

Reviewer #5: Yes

Reviewer #6: Yes

5. Is the manuscript presented in an intelligible fashion and written in standard English?

Reviewer #5: Yes

Reviewer #6: No

6. Review Comments to the Author

Reviewer #5: The article has improved substantially. However, there are still some details that must be corrected, namely:

* Whenever authors refer to the p-value, it must be in lower case, p instead of P. Authors should review the entire text.

* The titles of tables 2, 3, 4 and 5 are not correct, as they are not evaluating the form and intensity of the relationship between two variables, but the comparison of the values of a variable (which is quantitative) between the categories of a qualitative variable. Therefore, instead of correlation, there should be a comparison, for example comparison of the state of anxiety (assessed by STAI) between ...

Reviewer #6: It seems, the manuscript needs to rewrite or revise by an scientific writer for scientific article in order to be condense, clear methodology even the name of method " One hundred and thirty-three patients were

evaluated prospectively". Repeatable of the research work is important. Sampling and research community, response rate are important as well.

7. PLOS authors have the option to publish the peer review history of their article (what does this mean?). If published, this will include your full peer review and any attached files.

Reviewer #5: No

Reviewer #6: No

---

## [Author Response · Author response to Decision Letter 3]

29 Aug 2022

Reviewer #5

* The article has improved substantially. However, there are still some details that must be corrected, namely:

1. * Whenever authors refer to the p-value, it must be in lower case, p instead of P. Authors should review the entire text.

The authors are grateful for the observation made. The entire manuscript was revised and the p-values corrected.

2. * The titles of tables 2, 3, 4 and 5 are not correct, as they are not evaluating the form and intensity of the relationship between two variables, but the comparison of the values of a variable (which is quantitative) between the categories of a qualitative variable. Therefore, instead of correlation, there should be a comparison, for example comparison of the state of anxiety (assessed by STAI) between ...

The authors agree with the observations made. Conceptual changes were made to the table titles.

Reviewer #6

1. It seems, the manuscript needs to rewrite or revise by an scientific writer for scientific article in order to be condense, clear methodology even the name of method " One hundred and thirty-three patients were evaluated prospectively". Repeatable of the research work is important. Sampling and research community, response rate are important as well.

The authors are again grateful for the reviewer's notes. The entire article was reviewed and the necessary changes were made to clarify the scientific language.

---

## [Decision Letter · Decision Letter 4]

12 Sep 2022

Relationship Between Anxiety and Internet Searches Before Percutaneous Ultrasound-guided Diagnostic Procedures: A Prospective Cohort Study

PONE-D-21-27163R4

Dear Dr. Meira,

We’re pleased to inform you that your manuscript has been judged scientifically suitable for publication and will be formally accepted for publication once it meets all outstanding technical requirements.

Kind regards,

Francisco Sampaio, Ph.D.

Guest Editor

PLOS ONE

Additional Editor Comments (optional):

Reviewers' comments:

Reviewer's Responses to Questions

**Comments to the Author**

1. If the authors have adequately addressed your comments raised in a previous round of review and you feel that this manuscript is now acceptable for publication, you may indicate that here to bypass the “Comments to the Author” section, enter your conflict of interest statement in the “Confidential to Editor” section, and submit your "Accept" recommendation.

Reviewer #5: All comments have been addressed

2. Is the manuscript technically sound, and do the data support the conclusions?

Reviewer #5: Yes

3. Has the statistical analysis been performed appropriately and rigorously? 

Reviewer #5: Yes

4. Have the authors made all data underlying the findings in their manuscript fully available?

Reviewer #5: Yes

5. Is the manuscript presented in an intelligible fashion and written in standard English?

Reviewer #5: Yes

6. Review Comments to the Author

Reviewer #5: (No Response)

7. PLOS authors have the option to publish the peer review history of their article (what does this mean?). If published, this will include your full peer review and any attached files.

Reviewer #5: No

---

## [Editor Report · Acceptance letter]

26 Sep 2022

PONE-D-21-27163R4 

Relationship Between Anxiety and Internet Searches Before Percutaneous Ultrasound-guided Diagnostic Procedures: A Prospective Cohort Study 

Dear Dr. Meira:

I'm pleased to inform you that your manuscript has been deemed suitable for publication in PLOS ONE. Congratulations! Your manuscript is now with our production department. 

Kind regards, 

on behalf of

Professor Francisco Sampaio 

Guest Editor

PLOS ONE